# Bench-to-Bedside Studies of Arginine Deprivation in Cancer

**DOI:** 10.3390/molecules28052150

**Published:** 2023-02-24

**Authors:** George C. Field, Iuliia Pavlyk, Peter W. Szlosarek

**Affiliations:** 1Centre for Cancer Biomarkers and Biotherapeutics, Barts Cancer Institute, Queen Mary University of London, Charterhouse Square, London EC1M 6BQ, UK; 2Department of Medical Oncology, St. Bartholomew’s Hospital, Barts Health NHS Trust, West Smithfield, London EC1A 7BE, UK

**Keywords:** arginine, arginine deprivation, ADI-PEG20, argininosuccinate synthetase 1, monotherapy, combination therapy, cancer, biomarkers

## Abstract

Arginine is a semi-essential amino acid which becomes wholly essential in many cancers commonly due to the functional loss of Argininosuccinate Synthetase 1 (ASS1). As arginine is vital for a plethora of cellular processes, its deprivation provides a rationale strategy for combatting arginine-dependent cancers. Here we have focused on pegylated arginine deiminase (ADI-PEG20, pegargiminase)–mediated arginine deprivation therapy from preclinical through to clinical investigation, from monotherapy to combinations with other anticancer therapeutics. The translation of ADI-PEG20 from the first in vitro studies to the first positive phase 3 trial of arginine depletion in cancer is highlighted. Finally, this review discusses how the identification of biomarkers that may denote enhanced sensitivity to ADI-PEG20 beyond ASS1 may be realized in future clinical practice, thus personalising arginine deprivation therapy for patients with cancer.

## 1. Introduction

### 1.1. Role of Arginine in Cellular Function

Arginine is a semi-essential amino acid which is key in many neurological, immunological, and proliferative cellular functions [1]. As a precursor for a plethora of proteins, this dibasic amino acid also has critical roles in cellular metabolism [2]. Arginine is either sourced extracellularly, biosynthesised via the urea cycle, or generated by the arginine–citrulline cycle for nitric oxide (NO) synthesis [3].

The endogenous production of arginine in cells commences with ornithine, a precursor for polyamines that is generated within the mitochondria from glutamate (via glutamine) and proline or supplied from the urea cycle. Ornithine conversion to citrulline is the stage in the urea cycle in which ammonium (NH_4_^+^) is removed from the body. Carbamoyl phosphate synthetase is used to convert NH_4_^+^ and bicarbonate (HCO_3_^−^) into carbamoyl phosphate utilising 2 ATP molecules. Subsequently, ornithine transcarbamylase (OTC) converts ornithine and carbamoyl phosphate to citrulline, which is then transported into the cytosol. Citrulline and aspartate are now used to create arginine, catalysed by the enzymes argininosuccinate synthetase 1 (ASS1) and argininosuccinate lyase (ASL) and releasing fumarate in the process [4,5].

Arginine is also used in the production of creatine, vital for immune cell maintenance and thermogenesis [6], and agmatine, a metabolite possessing a variety of neurological and neuroprotective functions [7]. Since arginine is biosynthesised from internal metabolites in healthy cells, it is considered conditionally essential only during periods of injury and infection such as burns and sepsis, and intriguingly for many malignancies due in particular to dysregulation of the urea cycle enzyme, ASS1 [8].

This review focuses predominantly on preclinical and clinical research in which arginine-dependent (i.e., ASS1-deficient) malignancies have been targeted using pegylated arginine deiminase (ADI-PEG20, pegargiminase), either as monotherapy or in combination with other anticancer agents. The review will also comment on the current research expanding potential biomarkers in patients and their sensitivity to arginine-deprivation therapy.

### 1.2. Tumoral ASS1 Deficiency and Arginine Dependency

Despite healthy cells endogenously producing arginine for metabolic purposes, many cancers lose this ability for still incompletely understood reasons. Observed in 63% of patients with mesothelioma, reduced or total loss of ASS1 protein expression results in the cancer cells becoming wholly dependent on extracellular arginine for survival [9]. When investigating the effect of ASS1 deficiency in subjects with citrullinaemia type 1 and 2 (CTLN1/2), Rabinovich et al. [10] reported that decreased ASS1 expression led to increased proliferation in noncancerous cells, as a result of increased availability of aspartate for pyrimidine synthesis via carbamoyl phosphate synthetase 2, aspartate transcarbamylase, and dihydroorotase (CAD) complex activation [10]. To further validate their results in the context of cancer, Rabinovich et al. [10], investigated osteosarcoma cell lines either proficient (U2OS) or deficient (MNNG/HOS) in ASS1, showing a similar increase in proliferation with reduced ASS1 expression, associated with higher uracil levels and pyrimidine synthesis. Together, these data provide a key mechanistic explanation for other studies that reduced ASS1 protein expression drives and increases tumorigenesis and poor-prognosis cancers [11,12,13].

Notably, investigation into the epigenetic regulation of ASS1 found hypermethylation of the ASS1 promoter region in both solid and haematological malignancies with correspondingly low levels of ASS1 mRNA and protein [9,14]. While this correlation between promoter methylation and reduced ASS1 expression has now been reported more widely across a range of cancers [11,12,15,16], transcriptional roles have also been suggested for c-Myc and Sp4, as positive regulators, and HIF-1α, as a negative regulator, for arginine biosynthesis (Figure 1). Indeed, these studies emphasise that ASS1 expression and regulation is tissue-specific, with important implications for diagnosis and therapy [17,18]. More studies are needed to elucidate the complexity of the regulation of ASS1 across cancers.

As described above, ASS1 deficiency is linked in general to more aggressive cancer biology and worse clinical outcomes. ASS1 deficiency confers a critical dependency on extracellular arginine, and it is therefore vital to recognise how depletion of arginine may impact these ASS1 “low” cancers (Figure 2).

### 1.3. Arginine Deprivation: Preclinical Data

To gain a deeper understanding of how essential exogenous arginine is to ASS1 deficient cancers, it must first be depleted across a range of tumours. The arginine deiminase (ADI) pathway is found broadly across bacterial species, producing energy through the degradation of arginine and supporting survival in acidic environments through the production of ammonium [27]. The deimination of arginine to citrulline is catalysed by the enzyme ADI, followed by OTC conversion into carbamoyl phosphate and ornithine; then a final enzymatic reaction with carbamate kinase produces NH_4_^+^, CO_2_, and ATP [28]. On account of the short circulating half-life of ADI, ADI-PEG20 was developed as a 20 kDa PEGylated arginine deiminase to deplete arginine efficiently in vivo. Ensor et al. [29] were the first group to explore the effect of ADI-PEG20 in arginine auxotrophic malignancies. Having demonstrated that the PEGylation of ADI did not significantly alter the strong antitumour efficacy compared to that of native ADI in ASS1-deficient human melanoma and hepatocellular carcinoma (HCC) cell lines, Ensor et al. [29] explored how PEGylation optimised drug activity in vivo. Intramuscular injection of 5 IU/mouse of ADI-PEG20 revealed no detectable levels of arginine seven days post-injection, whereas administration of the same amount of native ADI saw plasma arginine levels drop by 40–50% for 24 h, returning to normal after 48 h. Furthermore, in vivo treatment of immunodeficient mice transplanted with human melanomas with native ADI revealed no difference in the survival rate compared to saline control. However, ADI-PEG2 treatment led to 50% of the mice surviving to 24 weeks compared to 7 weeks for the native ADI and control group mice [29]. Importantly, Ensor et al. [29] confirmed that ASS1 overexpression led to ADI-PEG20 resistance. These findings, also now recapitulated in various ASS1-deficient cancer cell lines [30,31,32,33], were the first to reveal the landscape of how ASS1 deficiency sensitises to ADI-PEG20 and, critically, how arginine deprivation therapy may be deployed against these malignancies.

ADI-PEG20 displays tumour cytotoxicity via a number of mechanisms, including caspase-dependent apoptosis of cancer cell lines [14,30,32,33,34]. However, others have reported that cell death is caspase-independent in prostate and small cell lung cancer (SCLC) cell lines and in glioblastoma cell lines [16,31,35]. Furthermore, when examining ASS1-deficient breast cancer cell lines (MDA-MB-231) supplemented with arginine-free media, Cheng et al. [36] identified that arginine starvation causes acetylation of glycolysis and mitochondrial ETC complex genes, resulting in chromatophagy [36]. Collectively, these studies illuminate the mechanisms that may underlie the cytotoxic effects of ADI-PEG20 with other therapeutics and how this may impact treatment according to tumour type.

Another method of arginine deprivation has employed arginases. First documented in the mid-1970s, rat or bovine arginases were cytotoxic for cultured lymphosarcoma cells [37]. However, bovine arginases were inactive when pegylated [38]. More recent reports revealed that recombinant human arginases (PEG-BCT-100; HuArgI(Co)PEG5000, PT01) possess significant in vivo antitumour activity preclinically [39,40,41]. This was also demonstrated in a phase I clinical trial, where PEG-BCT-100 plus oxaliplatin and capecitabine resulted in a 36% disease control rate (5 patients out 14), with one patient exhibiting a partial response [42]. However, due to limited clinical translational data from the bench to the clinic for recombinant human arginases [39,43,44], the predominant focus of this review is on ADI-PEG20, which has been translated from phase 1 to randomised phase 2 and phase 3 trials.

## 2. Monotherapy

Following on from the research by Ensor et al. [29], Izzo et al. [45] tested ADI-PEG20 clinically, demonstrating good tolerability and an early signal of activity in patients with HCC at the highest doses (≥160 IU/m^2^): two patients had a complete response, seven had a partial response, seven showed stable disease, and three had progressive disease [45]. Additional encouraging early-phase studies led to a global phase III trial in patients with HCC progressing on first-line sorafenib who were randomized to 18 mg/m^2^ (160 IU) ADI-PEG20 plus best supportive care (BSC) (n = 424) versus placebo plus BSC alone (n = 211) [46,47,48]. There was no significant difference in either the median overall survival (7.8 months vs 7.4 months, respectively) or the median progression-free survival (2.6 months in both groups). Post hoc analyses revealed that sorafenib is an inducer of ASS1 expression in HCC cell lines, and thus the study population was in all likelihood not enriched for patients with arginine auxotrophic HCC [48]. Thus, stratification by ASS1 status with pre-treatment HCC biopsies may be required going forward. Early resistance may also have accounted for the negative results with ASS1 re-expression documented in trials of ADI-PEG20 monotherapy and maintenance in patients with melanoma and thoracic cancers, respectively, and consistent with earlier preclinical data [17,49,50,51].

Concurrent with the trial development in HCC, ADI-PEG20 has been scrutinized for safety and efficacy in patients with advanced melanoma (including cutaneous and uveal disease) and acute myeloid leukemia (AML with 2 of 21 patients or 9.5% experiencing a complete response), and in the ADAM study exploring ADI-PEG20 (36 mg/m^2^) plus BSC versus BSC alone in pleural mesothelioma [49,52,53,54]. ADAM, being the only randomised ASS1-directed biomarker study of ADI-PEG20 monotherapy to date, provided a robust signal of increased progression-free survival (i.e., 3.2 versus 2.0 months), accompanied by a 10% risk of allergic/anaphylactoid reactions to the pegylated bacterial enzyme. Moreover, ASS1 was shown to be both predictive and prognostic, which is consistent with the preclinical data [9]. No complete or partial responses were seen by CT; however, a signal of disease stability was apparent, and partial metabolic responses by PET-CT were documented in almost 50% of patients receiving ADI-PEG20 [54].

To summarise the monotherapy experience, ADI-PEG20 was well-tolerated in general with few adverse effects linked to treatment but with variable efficacy across the study populations; this latter factor is likely due to differential enrichment for arginine-dependent cancers, to generation of resistance and, potentially, to different dosing schedules (for instance, 18 mg/m^2^–36 mg/m^2^ weekly across studies in melanoma). On the basis of the paradigm of L-asparaginase which was incorporated into the multimodality treatment of childhood leukaemia, further clinical studies of arginine deprivation should focus on combination drug approaches (summarised in Figure 3), posing the question: would ADI-PEG20 unleash its potential in this setting?

## 3. Combination Therapy

### 3.1. ADI-PEG20 and Antifolates

As discussed above, reduced expression of ASS1 contributes to a hyperproliferative phenotype through increased pyrimidine synthesis [10]. We studied the metabolic consequences of ASS1 epigenetic silencing in mesothelioma and bladder cancer cell lines, identifying reduced thymidine levels and increased levels of thymine and glutamine upon ADI-PEG20 treatment [12]. These changes were accompanied by reduced mRNA and protein expression of the folate-dependent nucleotide-synthesising enzymes thymidylate synthase (TS) and dihydrofolate reductase (DHFR) and suppressed ^3^H-thymidine uptake mediated by downregulation of thymidine kinase 1 in the cancer cell lines exposed to ADI-PEG20 [12]. Subsequently, we confirmed enhanced cytotoxicity of ADI-PEG20 with the antifolate chemotherapy drug pemetrexed, both in vitro and in vivo. This preclinical rationale was translated efficiently in the phase 1 TRAP clinical study programme assessing ADI-PEG20 with the pemetrexed–cisplatin doublet in a range of arginine-dependent cancers, including mesothelioma, non-small cell lung cancer, high-grade glioma, and uveal melanoma (ClinicalTrials.gov NCT02029690). Overall, these studies found good tolerability of the ADIPEMCIS drug triplet, with the majority of adverse effects being grade 1 or 2 (National Cancer Institute Common Terminology Criteria for Adverse Effects Version 4.03) and considered to be caused mainly by pemetrexed or cisplatin [51,55,56,57]. The minimum disease control rate was 70% (ranging from 70 to 100%), illustrated predominantly by stable disease, with some patients achieving partial response; complete responses were not seen.

In particular, an early robust signal of safety and activity was noted in patients with the most-aggressive non-epithelioid mesotheliomas (biphasic and sarcomatoid subtypes) leading to the initiation of the global randomised placebo-controlled phase 2/3 ATOMIC-meso study (ClinicalTrials.gov NCT02709512) [50,55]. Recently, ATOMIC-meso reported an increase in the median overall and progression-free survival in patients randomised to ADIPEMCIS, representing the first positive phase 3 study of amino acid deprivation since the development of asparaginase for the treatment of acute lymphoblastic leukaemia in the 1960s (manuscript in preparation) [58].

### 3.2. ADI-PEG20 and Gemcitabine

Gemcitabine, a nucleoside analog, is a key anticancer metabolite agent that interferes with DNA synthesis, in particular via inhibition of ribonucleotide reductase [59]. Gemcitabine is used in multiple cancer types, and resistance to it is an important clinical problem [60]. As ribonucleotide reductase subunit 2 (RRM2) converts gemcitabine from its active form into its inactive moiety, changes in this enzyme are thought to confer resistance and thus a target for enhancing the effects of gemcitabine [61]. During preclinical investigation, gemcitabine treatment of ASS1-deficient pancreatic cancer cell lines caused a dose-dependent increase in the expression of RRM2, which was then abrogated when combined with ADI-PEG20 [61]. Furthermore, the combination of ADI-PEG20 and gemcitabine showed increased antitumour efficacy in vivo over either therapeutic alone, illustrating its potential in the clinic [61].

Building on the clinical use of gemcitabine with docetaxel treatment [62] and the previous preclinical work of ADI-PEG20 with either gemcitabine [60] or docetaxel [34], Prudner et al. [59] sought to investigate the triplet combination of ADI-PEG20, gemcitabine, and docetaxel [60] in sarcoma cell lines. Here, the triplet combination was more cytotoxic compared to the gemcitabine and docetaxel doublet alone, promoting increased gemcitabine cellular uptake via increased hENT1 expression mediated by stabilisation and translocation of c-Myc to the nucleus [60]. A subsequent phase 2 clinical trial in soft-tissue sarcoma using ADI-PEG20, gemcitabine, and docetaxel has been completed, with preliminary data showing safety with a reduced dose of gemcitabine consistent with the mechanistic data, and a trend for greater benefit in patients with ASS1-negative tumours (ClinicalTrials.gov, NCT03449901 [63]); an additional study of this triplet is proceeding in relapsed non-small cell lung cancer and small cell lung cancer (ClinicalTrials.gov, NCT05616624). Further discussion of ADI-PEG20 and gemcitabine paired with another taxane, nab-paclitaxel, in pancreatic cancers is discussed in the next section [64].

### 3.3. ADI-PEG20 and Taxanes

Paclitaxel was the first cytotoxic taxane to enter the clinic, originally synthesised from the Pacific yew tree (*Taxus brevifolia*) and inducing cell cycle arrest via stabilization of microtubules resulting in apoptosis [65]. Taxanes may also play a role by impeding the anti-apoptotic protein BCL-2 and upregulating the tumour suppressor genes p53 and p21, hence their frequent application in cancer care [66]. Docetaxel is a semi-synthetic taxane, first approved in the mid-1990s for treatment of anthracycline-refractory metastatic breast cancer, and has now been used widely in prostate, ovarian, lung, and gastric cancers [67]. The preclinical combination of docetaxel and ADI-PEG20 in vivo led to a 75% decrease in prostate xenograft tumour size compared to mice treated with ADI-PEG20 alone [35].

To date, there are two published clinical trials investigating taxanes in combination with ADI-PEG20 [64,68]. Following on from the preclinical work by Kim et al. [35], the combination of ADI-PEG20 with docetaxel was tested in patients with advanced prostate cancer and non-small cell lung cancer. Separately, Lowery et al. [64] explored nab-paclitaxel and gemcitabine in patients with pancreatic cancer. While relatively high rates of grade 3–4 haematological toxicity were observed across both studies—not atypical for these regimens—the disease control rate of 77–94% is encouraging and merits further evaluation in larger studies [64,68,69].

### 3.4. ADI-PEG20 and Platinum

Cisplatin is another cytotoxic agent shown preclinically to enhance the effects of ADI-PEG20 by inhibition of DNA repair [70] and ASS1 suppression by upregulation of HIF-1α and downregulation of c-Myc in melanoma cell lines [71]. Yao et al. [72] tested the ADI-PEG20-doublet in a phase 1 clinical study across a range of ASS1-deficient multiple cancers including melanoma. The drug combination was safe, but limited activity was seen, with a partial response rate of 5% and a stable disease rate of 41%; a signal of activity was seen in uveal melanoma similar to that observed with single-agent ADI-PEG20 [53]. Since pemetrexed was approved as a cancer treatment in combination with cisplatin [73], and with rationale for arginine deprivation modulating folate-mediated one-carbon metabolism, the clinical studies of ADI-PEG20 plus pemetrexed and cisplatin (ADIPemCis) were initiated across multiple types of cancer. Excluding uveal melanoma, which is insensitive to antifolates and platinum, all of the studies exploring ADIPemCis reported consistently enhanced disease control rates over ADI-PEG20 monotherapy [50,51,55,56,57].

Considering 5-fluorouracil is an inhibitor of TS, an enzyme reduced by ADI-PEG20 treatment [12], and that platinum-based agents enhance the effects of ADI-PEG20 [70], the combination of mFOLFOX6 and ADI-PEG20 was assessed in HCC and other gastrointestinal malignancies [74]. In this trial, an overall response rate (ORR) of 9.3% was reported with a median PFS of 3.8 months, which was considered on par with historical controls; since ASS1 status pre-study entry was not mandated, this may explain the lack of perceived benefit [74]. Collectively, further biomarker studies may identify which patients would benefit, from deploying platinum in ADI-PEG20-containing chemo-regimens.

### 3.5. ADI-PEG20 and Doxorubicin

Doxorubicin is a member of a group of antibiotics known as anthracyclines; it intercalates with DNA, inhibits topoisomerase II, and triggers free radical–mediated cell death [75]. Qiu et al. [76] demonstrated a combination index (CI) < 1 and thus a synergistic effect of doxorubicin and ADI-PEG20 in the breast MDA-MB-231 cancer cell line, pointing towards the doublet’s potential in the clinic [76].

To this end, Yao et al. [77] conducted a phase 1 study of ADI-PEG20 in combination with liposomal doxorubicin for ASS1-negative metastatic solid cancers, focusing on HER2-negative breast cancer where anthracyclines have a key role [77]. The treatment was considered safe and tolerable. No objective drug responses were observed; however, 9/15 patients (60%) achieved stable disease and 3 patients experienced a minor reduction in tumour size [77]. Consistent with previous ADI-PEG20 studies, there was a reciprocal relationship between low peripheral blood arginine levels and high citrulline levels on study therapy [77]. Moreover, despite the levels of anti-ADI-PEG20 antibodies gradually increasing and then stabilising in 10 of the patients, in the other 5 patients there were no detectable antibody levels observed by week 8 [77]. The prolonged reduction in arginine levels is an outlier in the Yao et al. [77] study and may suggest an additional biological effect of the doxorubicin–ADI-PEG20 combination on ADI-PEG20 immunogenicity.

### 3.6. ADI-PEG20 and Cytarabine

Cytarabine is a deoxycytidine nucleoside analog which incorporates into the DNA, inhibiting replication and repair, and is a key cytotoxic agent in the management of haematological malignancies [78]. Previously, the combination of ADI-PEG20 and cytarabine was investigated in AML xenografted mice, where the doublet significantly reduced the percentage of AML cells in the bone marrow compared to either drug alone; the CI calculated at <1, indicative of drug synergy [79].

Compared with the monotherapy experience in AML described earlier, of the 18 patients with evaluable disease, 7 achieved complete remission (38.9%), 1 had a partial response (5.6%), and 6 had stable disease (33.3%), denoting a disease control rate of 77.8% [80]. Thus, the combination treatment exceeded the monotherapy efficacy almost 4-fold and was well tolerated by the patients [71]. These results, although encouraging, are based on small numbers; a larger, placebo-controlled study would be justified. In particular, ADI-PEG20 and low-dose cytarabine may be a useful regimen in combatting haematological malignancies in those not fit for more intensive chemotherapies or hematopoietic stem cell transplants [71].

### 3.7. ADI-PEG20 and Immunotherapy

Brin et al. [81] identified a rationale for combining anti-PD-1 monoclonal antibodies and ADI-PEG20 in the B16-F10 melanoma and CT26 colorectal cancer models, with evidence of enhanced disease control with induction of tumoural PD-L1 expression and T cell infiltration [81]. Clinically, rebiopsy studies showed increased tumoural PD-L1 expression and T cell aggregates on disease progression in patients with mesothelioma treated with maintenance ADI-PEG20 on the TRAP programme [50]. These data and the known recycling of the ADI-PEG20 degradation product, citrulline to arginine by T cells, led to the development of the first phase 1 immunometabolic study of arginine deprivation with immune checkpoint blockade. Combination treatment with ADI-PEG20 and the PD-1 antagonist pembrolizumab was well-tolerated overall; however, there was an interesting grade 3–4 neutropenia rate of 40%, significantly higher than that expected with either drug alone (10/25 patients) [82]. Six of the 25 (24%) enrolled patients achieved a partial response, and 7 experienced stable disease with a disease control rate of 52%. Additionally, of the evaluable patients (23/25), the median overall survival was 8.5 months and the progression free survival was 1.9 months. Blood arginine levels declined to undetectable levels between weeks 1 and 3 and then gradually rose to plateau at around 50% for the duration of the study, consistent with steady increases in the levels of anti-ADI-PEG20 antibodies [81]. Notably, of 12 available paired biopsies, 10 showed significant increases in the level of CD3^+^ T cells posttreatment, with all 6 responding patients in the dose expansion cohort presenting low (2/6) or absent (4/6) baseline levels of PD-L1. Moreover, paired tumour biopsies revealed an increase in PD-L1 levels in 3 out 10 patients from the maximum tolerated dose cohort [81].

Conversely, when exploring the triplet therapy of nivolumab, ipilimumab, and ADI-PEG20 in patients with uveal melanoma, Kraehenbuehl et al. [83] reported less-encouraging results based on a small dataset [74]. Although the triplet combination was deemed safe and tolerable, out of 9 patients, none experienced complete or partial responses, only 2 showed stable disease, and all displayed progressive disease resulting in a median progression-free survival of 1.5 months and a median overall survival of 8.6 months [74]. Paired biopsies available in all the patients found no significant differences in the levels of either PD-L1^+^ or CD8^+^ macrophages between pre- and posttreatment, but it was noted that the percentage of PD-L1^+^ cells did vary considerably in the untreated patients [74]. Finally, considering that interferon gamma (IFN-γ) induces expression of PD-L1, the increase in the expression of IFN-γ-dependent genes in patients who had received ICB therapy prior to entering this trial compared to those who had not may explain the variation in PD-L1^+^ cells in the untreated patients [74].

While both the doublet and triplet immunometabolic studies are too small to make definitive conclusions, and there may be an earlier increase in anti-ADI-PEG20 antibodies with ipilimumab therapy, further investigation of the immunometabolic rationale is underway in the context of CAR-T cells [82,83,84].

## 4. ADI-PEG20 Resistance

Resistance to ADI-PEG20 is likely multifactorial and dependent on tumour type. Broadly, several mechanisms have been postulated or identified. From a drug-intrinsic perspective, anti-ADI-PEG20 antibodies increase with treatment, and while this leads to plasma arginine elevation over time, Yao et al. [77] showed that 4 h post-dose, ADI-PEG20 maintains low arginine levels. From the perspective of the tumour cell, resistance may be mediated via ASS1 upregulation, autophagy, and stromal cell refueling of arginine or its precursor argininosuccinate [50]. Moreover, in the TRAP dose-expansion cohort of patients with NSCLC, we pursued a pharmacogenomic approach to correlate drug sensitivity/resistance to ADI-PEG20 with next-generation sequencing (NGS). We identified several molecular markers, with a doubling in the expected rate of P53 mutations (64.7%), while KRAS mutations were at the upper expected limit for non-squamous NSCLC (35.3%). Further work is needed to clarify how P53 mutations modulate sensitivity/resistance to ADI-PEG20 [85].

## 5. Biomarkers and Arginine Deprivation Treatment

Following on from the negative phase 3 HCC study, Chu et al. [86] pursued comprehensive analysis of single nucleotide polymorphisms (SNPs) in ADI-PEG20-treated, advanced patients with liver cancer to identify potential biomarkers of response [48,86]. Having previously identified SNPs that may be related to therapeutic outcome [87,88], Chu et al. [86] set out to investigate these SNPs in patients treated with ADI-PEG20 (cohort 1: 113 from the trial conducted by Abou-Alfa et al. [48] and cohort 2: 47 from the trial by Harding et al. [74]. Cohort 2 was also treated with mFOLFOX6. Interestingly, in cohort 1 the *WOXX-*rs13338697-GG genotype was associated with improved overall survival and *WOXX-*rs6025211-TT with reduced time to tumour progression (TTP); no genotype was associated with any outcome in cohort 2 [86].

Moreover, Western blot and immunohistochemical analyses showed that the *WOXX-*rs13338697-GG genotype correlated with a reduction in the ratio of WWOX expression in tumour to non-tumour (T/N) tissue compared to other genotypes GT and TT, and a significant positive correlation between WWOX T/N and ASS1 T/N expression [86]. As loss of function in *WOXX* results in an increase in HIF-1α in HCC [89], and increased HIF-1α expression decreases ASS1 expression [71,90], Chu et al. [86] showed that silencing *WOXX* (with CoCl_2_) causes an increase in HIF-1α and a decrease in ASS1 expression and consequently sensitises cells to ADI-PEG20 treatment [86]. On the basis of this rationale, a randomised placebo-controlled trial is underway exploring the potential of WWOX-GG in selecting patients with HCC for ADI-PEG20 monotherapy (ClinicalTrials.gov Identifier: NCT05317819).

Another recent study by Barnett et al. [91] has identified BAP1 loss as a regulator of high ASS1 expression in mesothelioma, opening up the possibility for further personalization of ADI-PEG20 in patients with epithelioid disease, namely those with BAP1-proficient disease (and low ASS1) which heralds a poorer prognosis similar to non-epithelioid disease (which is largely BAP1-retained). These reports highlight how the recognition of specific biomarkers may lead to further optimization of arginine deprivation for patients with treatment-refractory malignances (Figure 4).

## 6. Conclusions

Moving stepwise from monotherapy to combination regimens, informed by preclinical mechanistic data, has been critical in developing ADI-PEG20 for the treatment of cancer, spearheaded in mesothelioma. This approach has succeeded on the basis of the asparaginase template whereby asparagine deprivation for acute lymphoblastic leukaemia is deployed in the context of multimodality therapy rather than as monotherapy. Nonetheless, with increasing knowledge of tumoural arginine dependency, optimal targeting by ADI-PEG20 is likely to come via increased personalization, whether tumour histology–based or via the use of specific molecular biomarkers. Strategies to circumvent resistance to ADI-PEG20 will also be needed to realise the full potential of arginine deprivation for the treatment of cancer.

## Figures and Tables

**Figure 1 molecules-28-02150-f001:**
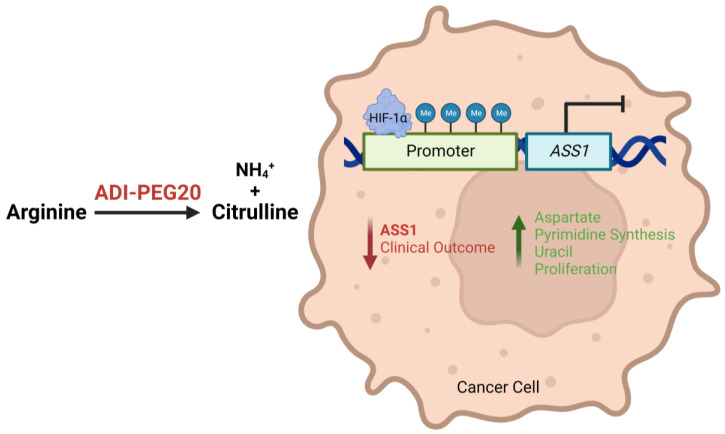
Illustration describing how epigenetic and transcriptional loss of ASS1 expression leads to a more aggressive tumour phenotype and worse patient outcomes, and how ADI-PEG20 may target ASS1-deficient and arginine-dependent tumours by depleting exogenous arginine.

**Figure 2 molecules-28-02150-f002:**
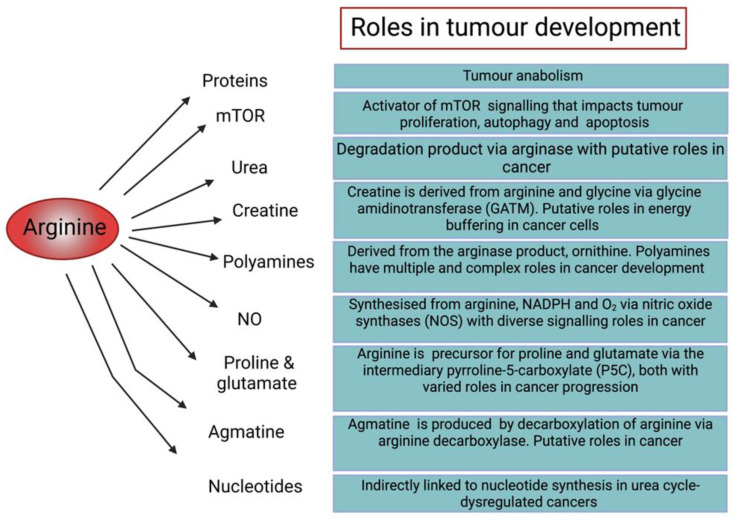
Diagram illustrating the roles of arginine in cellular function and tumour development [6,19,20,21,22,23,24,25,26].

**Figure 3 molecules-28-02150-f003:**
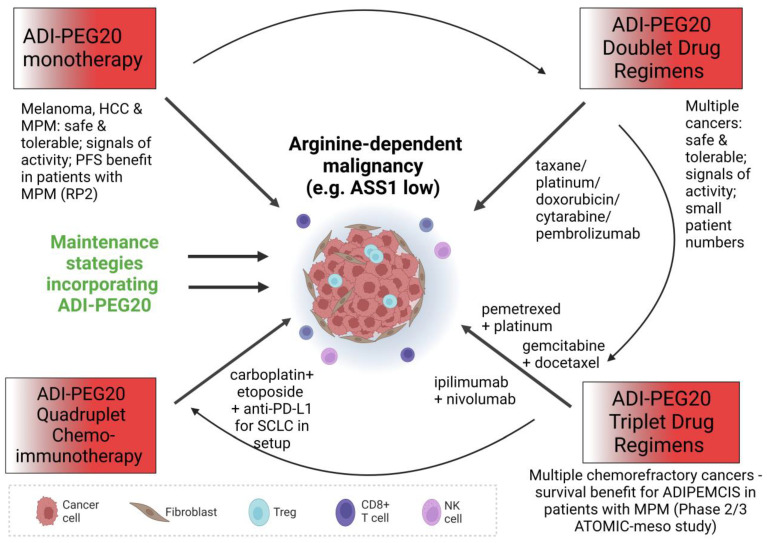
Diagram illustrating the different modes of attack of ADI-PEG20 in monotherapy and in combination with various other anticancer treatment options.

**Figure 4 molecules-28-02150-f004:**
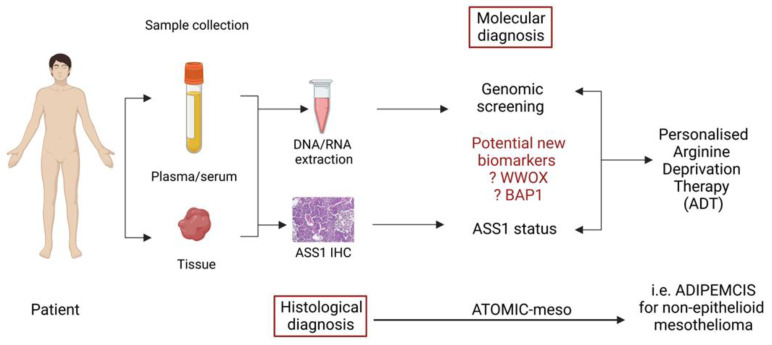
Schematic diagram suggesting how identification of biomarkers from previous ADI-PEG20 treated patients may help define personalised arginine deprivation therapy.

## Data Availability

Not applicable.

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
