# Peer review of "Bench-to-Bedside Studies of Arginine Deprivation in Cancer"

_molecules, 2023, doi:10.3390/molecules28052150_

Round 1
Reviewer 1 Report
nice to discuss further why some clinical trials were positive for ADI PEG 20 and others were not. For example, the melanoma phase II trial at Sloan Kettering used 1/2 recommended dose of the drug compared to the trial at Sylvester Cancer Center. Blood level of arginine may not reflect tissue level also. Also may be necessary to hit the tumor hard with argininie depletion before antibodies have a chance to form.
The phase III trial in HCC, did not stratify patients according to tumor ASS expression and may explain the negative results since tumors that were ASS positive were included in the analysis since no tumor biopsies were done for evaluation of ASS expression.
Author Response
We are grateful to the Reviewers for their helpful comments and have addressed these as follows:
Reviewer 1
We concur with the comments and have included a statement for the variable trial results of ADI-PEG20 in melanoma as a possible consequence of drug dosage and the negative HCC results due to lack of stratifaction by ASS1 status. We have included the following sentences to satisfy the revision:
- Thus, stratification by ASS1 status with pre-treatment HCC biopsies may be required going forward.
- and potentially different dosing schedules (for instance, 18mg/m2 - 36mg/m2 weekly across studies in melanoma).
Reviewer 2 Report
This review summarizes the recent developments of cancer therapeutics by depriving arginine. The manuscript contains cutting-edge knowledges regarding the potential for the clinical values of arginine depletion on cancer therapies, clinical trials of ADI-PEG20 treatment for cancer therapy and its future perspective. The document is well-written on the whole. I have some suggestions.
1. line10-11
There is a repetition of same sentence. I suggest the authors should have the manuscript checked carefully before submission.
2. The manuscript would be more understandable if the authors could add one simple figure to show the mechanism of down-regulation of ASS1 in cancer (epigenetics, transcription factor etc.), how decreased ASS1 rises the aggressiveness of tumors, and the mechanism of anticancer action of ADI-PEG20.
Author Response
We are grateful to the Reviewers for their helpful comments and have addressed these as follows:
Reviewer 2
As advised, we have provided a new figure 1 in the manuscript summarising the main mechanisms underlying ASS1 down-regulation in cancer (i.e. epigenetics, transcription factors) and how decreased ASS1 links to the aggressiveness of tumours, and sensitivity to the anticancer action of ADI-PEG20. The subsequent figures have been relabelled accordingly (Figs 2-4). To the best of our knowledge, repetition within the manuscript has been removed, the grammar has been improved and typos have been corrected.